# Characterisation of *Trichomonas vaginalis* Isolates Collected from Patients in Vienna between 2019 and 2021

**DOI:** 10.3390/ijms232012422

**Published:** 2022-10-17

**Authors:** Iwona Lesiak-Markowicz, Julia Walochnik, Angelika Stary, Ursula Fürnkranz

**Affiliations:** 1Intitute for Specific Prophylaxis and Tropical Medicine (ISPTM), Centre for Pathophysiology, Infectiology and Immunology, Medical University of Vienna, 1090 Vienna, Austria; 2Pilzambulatorium Schloesselgasse, Oupatients Centre for Diagnosis of Venero-Dermatological Diseases, 1080 Vienna, Austria

**Keywords:** *Trichomonas vaginalis*, *Trichomonas vaginalis* virus, *Mycoplasma hominis*, endosymbionts

## Abstract

*Trichomonas vaginalis* (TV) is the causative agent of trichomoniasis, the most common nonviral sexually transmitted disease. TV can carry symbionts such as *Trichomonas vaginalis* virus (TVV) or *Mycoplasma hominis*. Four distinct strains of TV are known: TVV1, TVV2, TVV3, and TVV4. The aim of the current study was to characterise TV isolates from Austrian patients for the presence of symbionts, and to determine their effect on metronidazole susceptibility and cytotoxicity against HeLa cells. We collected 82 TV isolates and detected presence of TVV (TVV1, TVV2, or TVV3) in 29 of them (35%); no TVV4 was detected. *M. hominis* was detected in vaginal/urethral swabs by culture in 37% of the TV-positive patients; *M. hominis* DNA was found in 28% of the TV isolates by PCR. In 15% of the patients, *M. hominis* was detected in the clinical samples as well as within the respective TV isolates. In 22% of the patients, *M. hominis* was detected by culture only. In 11 patients, *M. hominis* was detected only within the respective cultured TV isolates (13%), while the swab samples were negative for *M. hominis*. Our results provide a first insight into the distribution of symbionts in TV isolates from Austrian patients. We did not observe significant effects of the symbionts on metronidazole susceptibility, cytotoxicity, or severity of symptoms.

## 1. Introduction

Trichomoniasis is the most common nonviral sexually transmitted infection (STI) of the human urogenital tract, with 156 million new cases estimated in 2016 [1]. The global prevalence of TV has been reported to be around 5.0% for women and 0.6% for men [1]. The majority of women and men infected with TV reportedly remain asymptomatic; however, in one-third of women a latent period of up to 6 months with subsequent appearance of symptoms has been described [2,3,4]. Trichomoniasis symptoms include vaginitis in women (vaginal erythema, itching, vaginal discharge, odour, or dysuria) and urethritis in men. Commonly known sequelae include adverse pregnancy outcomes and an increased risk of the acquisition and transmission of HIV and HPV [5,6]. *Trichomonas vaginalis* virus (TVV) is a nonsegmented, 4.5–5 kbp, double-stranded RNA (dsRNA) virus belonging to the Totiviridae family, members of which infect fungi [7] and a variety of parasitic protists, including *Giardia duodenalis* [8] and *Leishmania braziliensis* [9]. TVV is recognised as the first dsRNA virus observed in a protozoan parasite [10] and has been divided into four subspecies: TVV1, TVV2, TVV3, and TVV4 [11]. Different TVV subspecies can coinfect and coexist inside the same TV isolate [12]. Among the four distinct species of TVV, TVV1 is the most prevalent, followed by TVV2, TVV3, and TVV4 [11]. It has been postulated that TVVs may upregulate the expression of immunogenic surface proteins and cysteine proteinases [13]; however, the clinical significance of TVVs and their effects on the pathogenicity of TV is still not well-known. Graves et al. [11] found no association of TVV positive (TVV+) isolates with clinical symptoms, reinfections, or metronidazole resistance. *Mycoplasma* spp. are the smallest free-living, self-replicating, cell-wall-deficient bacteria known so far [14,15]. *Mycoplasma* spp. are typically extracellular and surface-attached; however, a few species can enter host cells and establish an intracellular niche [16]. TV is known to act as a niche and vector for *M. hominis* [17]. *M. hominis* is the most frequently isolated microorganism from the genital tract of both men and women, found as part of the natural flora but also considered as a putative pathogen responsible for a variety of urogenital infections, neonatal infections, and systemic infections in immunocompromised patients [18]. 5-nitroimidazoles, especially metronidazole, are the most widely used antimicrobial agents for the treatment of trichomoniasis. Metronidazole, developed in the 1950s and approved for the treatment of trichomoniasis in the early 1960s, was the first drug to have a cure rate approaching 100% when used systemically [19]. Some cases of drug resistance and treatment failure of metronidazole have been observed since 1962 [20], but resistance remained rare and could mostly be overcome with larger doses of the drug. Resistance mechanisms to metronidazole have been studied by incubating TV strains with different drug concentrations under aerobic and anaerobic conditions in vitro; however, the underlying mechanisms of resistance are still not fully understood. The current treatment guidelines recommend 500 mg metronidazole twice daily for 7 days [21]. The aim of this retrospective study was to characterise TV isolates collected from patients visiting the Outpatients Centre for the Diagnosis of Venero-Dermatological Diseases in Vienna with regard to their symbionts, drug susceptibilities to metronidazole, and cytotoxicities to human cells.

## 2. Results

### 2.1. Collection and Characteristic of TV Isolates

TV isolates were collected from 82 patients between 2019 and 2021: 10 isolates derived from men (mean age 46) and 72 from women (mean age 41.2). Thirteen patients did not report any clinical symptoms (7 women, 6 men); in four patients (only female), TV was the only pathogen detected (Table 1), and these patients reported clinical symptoms including pain, itching, discharge, and urinary tract infection symptoms. In the other 78 patients attending the Outpatients Centre, other potential pathogens such as *Gardnerella vaginalis*, *Prevotella* spp., *Ureaplasma* spp., *Mycoplasma genitalium*, and *Chlamydia trachomatis*, as well as *Candida* spp. were also detected apart from TV.

### 2.2. Detection of TVVs in Clinical Isolates

In 29/82 (35%) TV samples, TVVs were detected; 16 of these 29 (55%) were identified as TVV1, 2/29 (7%) as TVV2, and 5/29 (17%) as TVV3; no TVV4 was detected in our collection of TV isolates. A total of 4/29 isolates (14%) harboured both TVV1 and TVV2, and 2/29 (7%) were positive for TVV1 and TVV3. In 53/82 (65%) no TVVs were detected.

### 2.3. Prevalence of TVVs in Symptomatic and Asymptomatic Patients

We correlated the occurrence of symptoms as reported by the patients infected with TV with the presence of different TVVs (Figure 1). Twenty patients positive for TVV (20/29 76%) reported symptoms (e.g., vaginitis, pruritus, pain). Out of the 13 TV isolates obtained from patients without reported symptoms, 6 isolates (6/13 46%) revealed the presence of TVV1; in the other 7 isolates (7/13 54%), no TVVs were detected. Among four TV isolates from patients with TV only (no other pathogen detected), TVVs were detected in three isolates; each TV isolate harboured one TVV: TVV1, TVV2, or TVV3. The patient with TV + TVV1 had not complained about any symptoms.

### 2.4. Detection of M. hominis in TV and Vaginal/Urethral Swabs

*M. hominis* was detected in vaginal/urethral swabs (culture method: colonies on an agar plate in the Outpatients Centre) in 30 of the 82 TV-positive patients (37%) (27 female, 3 male); *M. hominis* DNA was proven in 23 of the 82 cultured TV isolates by PCR (28%) (Table 2). In 12 patients (12/82; 15%), with *M. hominis* detected by culture in the Outpatients Centre, *M. hominis* DNA was also detected inside TV. In 18 patients (18/82; 22%), *M. hominis* was detected only by culture, not inside TV by PCR. In 11 cultured TV isolates (11/82; 13%) *M. hominis* was only detected by PCR (Table 2). 

### 2.5. Prevalence of M. hominis in Symptomatic and Asymptomatic Patients

Among the 30 patients with *M. hominis* detected in vaginal/urethral swabs by the culture method, 4 patients (2 female, 2 male) did not report any symptoms (13%). In the group of 23 patients, where *M. hominis* was detected inside TV by PCR, 4 patients (1 female, 3 male) were asymptomatic (17%). In the group of 12 patients where *M. hominis* was confirmed by culture as well as inside TV by PCR, 2 male patients (17%) were free of symptoms. In the group of 18 patients positive for *M. hominis* by culture but negative by PCR (not inside TV), 2 female patients (11%) were without any symptoms; and of the 11 patients who were TV-positive for *M. hominis* by PCR but negative by culture, 1 male and 1 female patient (18%) were asymptomatic. The remaining patients reported symptoms as mentioned above.

### 2.6. MICs

TV isolates were screened for metronidazole susceptibility under aerobic and anaerobic conditions in parallel. After 48 h incubation in anaerobic conditions, all tested isolates showed metronidazole sensitivity, with MICs ranging from 0.5 to 2 µg/mL. Under aerobic conditions, MICs of metronidazole ranged between 0.5 and 32 µg/mL (Table 3). We did not observe a discrepancy between MIC and MLC values, and our results did not show significant differences in the MICs in correlation to the symbionts of the respective TV isolates.

### 2.7. LDH Assay

Cytotoxicity of TVs was tested for four different TV groups (five isolates per group). Cytotoxicity of TVs without symbionts ranged from 0.85% to 9.17%, while TVs with TVVs showed cytotoxicities between 1.88% and 11.29%, TVs with *M. hominis* between 0.81% and 4.48%, and TVs with both symbionts between 0.46% and 8.02% (Figure 2). No statistically significant differences in cytotoxicity between the investigated groups of isolates were found. Microscopic examination revealed no differences in monolayer disruption between TVs with or without symbionts after two hours’ incubation of HeLa cells with the respective TVs.

## 3. Discussion

We detected TVVs in 35% (29/82) and *M. hominis* in 28% (23/82) of the collected TV isolates. To the best of our knowledge, the current study is the first reporting the prevalence of symbionts in *T. vaginalis* isolated from STI patients in Austria. In other European countries, TVV prevalence ranges from 33.3% to 50.4% (Czech Republic: 33–100%; Italy: 50%; the Netherlands: 50.4%) [11,22]. Masha et al. [23] investigated the prevalence of TVV subspecies in Kenya, showing that TVV1 was the most prevalent (39.1%), followed by TVV2 (26.1%), TVV3 (17.4%), and TVV4 (13%). In Brazil, TVV1 was also the most prevalent TVV species (77.5%), followed by TVV2 (30%), TVV3 (30%), and TVV4 (10%) [24]. In contrast, in Italy, the most prevalent virus was TVV2 (79.17%), followed by TVV1 and TVV3 [22]. Our data, indicating that TVV1 was the most prevalent TVV species in Austrian *T. vaginalis* isolates, fit into the global picture of TVV positivity in TV.

TVV prevalence and its clinical implications are still poorly understood. It has been shown that the presence of TVV can significantly enhance proinflammatory responses to TV in human epithelial culture models [25]. Double-stranded RNA viruses in other protozoan parasites (*Leishmania* spp., *Giardia* spp., *Plasmodium* spp., *Entamoeba* spp., *Babesia* spp.) are believed to be involved in disease outcomes. It has been demonstrated that differences in clinical manifestations between different *Leishmania* strains correlate to the presence of *Leishmania* virus (LRV), although this was not the only factor involved [26]. Moreover, *L. braziliensis guyanensis* infected with LRV1 induced more metastasising lesions in mice than an LRV-free strain [27]. Moreover, TVVs may upregulate the expression of immunogenic surface proteins, such as P270, and can cause phenotypic changes that may affect *T. vaginalis* virulence [13]. We could observe that cytotoxicity of TV isolates harbouring TVV or TVV + *M. hominis* was enhanced; albeit this was not statistically significant. Fraga et al. [28] reported that TV isolates infected with TVV showed increased cytoadhesion to HeLa cells. Cytotoxicity of TV is a contact-dependent mechanism. It was shown in 1984 by Alderete and Pearlman [29] that exposure of monolayer cultures of human cervical cancer (HeLa), human epithelial (HEp-2), normal baboon testicular (NBT), and monkey kidney (Vero) cells to live pathogenic *T. vaginalis* led to extensive disruption of the monolayers. Menezes et al. [30] observed that the LDH release from HVECs (human vaginal epithelial cells) varied between 5% and 74%, and this correlated to distinct TV strains’ cytolysis profiles. Analysis of the different groups of TV isolates (with and without symbionts) in the current study did not reveal microscopically detectable differences in the adhesion to or disruption of monolayers of HeLa cells after 2 h incubation and showed low levels of cytotoxicity. Rassmusen et al. [31] used human vaginal epithelial cells and showed that a strain of TV freshly isolated from a patient with severe symptoms caused extensive disruption of the epithelial cell monolayer, whereas a strain maintained in axenic cultures for several years did not. It has been argued that parasitic organisms maintained in culture for a long time lose their potential to infect host cells [32]. However, the observed low cytotoxicity in our study cannot be due to long culture periods, as the isolates were in culture only for several days before the experiments.

In four female patients of the current study, no other infectious agent than TV was detected; thus, we chose these patients to investigate a possible connection between symbionts and symptoms. In three of these TV isolates, TVV presence was proven; no *M. hominis* was detected. The patient with TV harbouring TVV1 did not complain about any symptoms, whereas the patient with TVV2 reported vaginal discharge and pruritus and the patient with TVV3 complained about pain and vaginal discharge. Among the 13 TV isolates collected from asymptomatic patients, only the presence of TVV1 was confirmed in 6 isolates; the other isolates were TVV-free. Fichorova et al. [25] reported that infections of TV with TVV1 were associated with mild clinical symptoms, whereas the presence of TVV2 inside TV was associated with severe symptoms. Thus, the assumption that the presence of TVV1 is not associated with severe symptoms was partly confirmed in our study, although the sample size was quite small. However, these data are difficult to compare, as at the time when the experiments were performed, it was not possible to clear TVVs from TV, and thus it was not possible to compare the same TV isolate with and without TVV. Recently, a cytidine nucleoside analogue has been shown to be an effective antiviral drug against TVV [33]. This surely is a major advance and will allow future studies to investigate the role of individual TVVs on pathogenicity, as well as metronidazole susceptibility.

Metronidazole has remained a highly effective drug to treat trichomoniasis until now, and drug resistance is still rare [34]. However, in some parts of the world, such as Papua New Guinea, resistance rates can exceed 15% [35]. Metronidazole resistance in TV can be either “aerobic“ (clinical) or “anaerobic“ (laboratory-induced) [36,37] with resistance being defined as >100 µg/mL and >3.1 µg/mL, respectively [38]. Our results revealed no association between TVV presence and metronidazole susceptibility, or resistance. Malla et al. [39], as well as Snipes et al. [40] showed that *Trichomonas* carrying TVVs showed increased susceptibility to metronidazole. However, Graves et al. [11] detected no association of TVV positivity and metronidazole susceptibility and suggested that TVV may be commensal to TV. In case of *Giardia duodenalis*, there was also no evidence that infection with dsRNA virus impacts resistance to metronidazole, which is also the first-line drug against *Giardia* [41].

The relationship between TV and *Mycoplasma* *hominis* has been investigated in different studies, indicating that the majority of women positive for TV also carry *M. hominis* [42]. Ioannidis et al. also showed that *Ureaplasma* has a tendence to be intracellular in TV [43]. A total of 37% of the patients enrolled in our study were *M. hominis*-positive. In a study performed in Italy, 78.6% of women with trichomoniasis were positive for *M. hominis* [44], and in the Netherlands, this figure was 79% [45]. Moreover, TV acts as a potential carrier for *M. hominis* [46], and the percentage of TV infected with *M. hominis* ranges from 5 to 89% depending on the geographic origin [24,28,47]. We detected *M. hominis* inside and/or membrane-associated to TV in 28% of the TV isolates. When *M. hominis* is hosted by TV, it benefits not only by evading the host immune response but can possibly also hide from antibiotics, which are unable to cross the cell membrane of TV. *M. hominis* is resistant to anti-*Trichomonas* drugs such as metronidazole and tinidazole [48,49]. Xiao et al. [50] suggested that *M. hominis* may confer (or contribute to) drug resistance in TV but to a varying degree, because some infected isolates were not demonstrably metronidazole-resistant when assessed in vitro. In Fürnkranz et al. [51] artificially *M. hominis*-infected TV strains showed 2 times higher MIC to metronidazole than the respective noninfected strains; moreover, genes involved in metronidazole resistance were influenced by the presence of *M. hominis*. On the other hand, Margarita et al. [52] showed that the presence of *M. hominis* significantly increases the sensitivity to metronidazole in TV and affects gene expression. Thus, *M. hominis* might have an impact on drug susceptibility of TV. It was also shown that the microbial association with *M. hominis,* as well as with Ca. M. girerdii, could modulate the virulence of TV [53]. Our recent results did not confirm this; however, we did not clear the TVs from *M. hominis* to test the same isolate with and without symbionts. Further studies would be necessary to investigate if the same TV strains with and without *M. hominis* show differences, and if it is possible to infect TVV-free TVs with TVV and then reveal specific impacts of the respective TVVs.

After TV treatment with metronidazole, *M. hominis* can be released and cause invasion not only into the foetal membranes but also into amniotic fluid, resulting in adverse pregnancy outcomes [50]. However, we must emphasise that *T. vaginalis* infection is not always associated with the presence of *M. hominis* and vice versa. In the current study, the majority of *T. vaginalis*-positive patients with *M. hominis* infection confirmed either by culture, culture and PCR (i.e., intracellular/membrane associated *M. hominis*), or by PCR only had symptoms such as pruritus, vaginal discharge, pain, bacterial vaginosis, etc. Plummer et al. [54] revealed that *M. hominis* was associated with symptoms/signs in cases of bacterial vaginosis (BV); but not associated with symptoms/signs in women without BV. On the other hand, during one year of sampling of *M. hominis* at the Outpatient Centre, we confirmed 56 patients (31 men and 25 women) where *M. hominis* was the only infectious agent found. Of these, 14 men and 20 women (45% and 80%, respectively) reported symptoms, thus leading to the assumption that *M. hominis* was the causative agent.

In any case, it is still controversially discussed, if *M. hominis* should be considered a pathogen at all. Arya et al. [55] did not find evidence that *M. hominis* is a vaginal pathogen in adults, and the European Academy of Dermatology and Venereology recommended in 2018 that patients should not even be screened for the presence of *M. hominis* [56]. On the other hand, the high frequency of *M. hominis* in prostate cancer patients indicates a yet hidden role of these bacteria in the development of prostate cancer during chronic, silent, and asymptomatic colonisation of the prostate [57]. Ten patients in our study would have remained undiagnosed for *M. hominis* because of its exclusive presence inside TV. Thus, if a screening for *M. hominis* should be implemented, it might also be essential to screen TV for *M. hominis*.

In conclusion, our results provide a first insight into the distribution of symbionts (TVVs, *M. hominis*) in TV isolates from Austrian patients. We detected the presence of TVV1, TVV2, and TVV3; no TVV4 was detected in our collection. *M. hominis* was detected in vaginal/urethral swabs by the culture method (i.e., in the patient), as well as by PCR (i.e., intracellular and, or membrane associated in TV). We did not observe significant effects of the symbionts on metronidazole susceptibility, cytotoxicity, or severity of symptoms reported by the patients.

## 4. Materials and Methods

### 4.1. Collection and Culture of Trichomonas vaginalis Isolates

Vaginal/urethral swabs were collected from symptomatic and asymptomatic patients attending the Outpatients Centre for the Diagnosis of Infectious Venero-Dermatological Diseases in Vienna between 2019 and 2021. Smears of the swabs were cultured on specific *Trichomonas*-culture plates [58] for 5 days and examined microscopically for the presence of mobile TV. Positive cultures were transported to the Institute of Specific Prophylaxis and Tropical Medicine (ISPTM); parasites were transferred from the plate into liquid TYM Medium (Trypticase-Pepton Medium) [59] supplemented with 10% heat-inactivated horse serum and 1 g cysteine per litre (pH 6.0), and cultured microaerobically in culture flasks at 37 °C. All patients’ data except for symptoms, coinfections, and gender were blinded. As the study was retrospective, no consent was needed from the participants.

### 4.2. RNA Extraction and First-Strand cDNA Synthesis

In order to detect TVV, whole-cell RNA was isolated. Briefly, TV cultures were washed twice with 1× PBS (centrifugation at 700× *g* for 10 min). Afterwards, nucleic acid extraction was performed using the GeneJET RNA Purification Kit (Thermo Fisher Scientific, Vilnius, Lithuania) following the manufacturer’s instructions. The isolated RNA was treated with DNase (Roche, Basel, Switzerland), in order to ensure the absence of genomic DNA. First-strand cDNA was synthesised using RevertAid First Strand cDNA Synthesis Kit (Thermo Fisher Scientific-Life Technologies); 1 µg of RNA was used as template for cDNA synthesis.

### 4.3. PCR for TVVs

cDNAs were used for PCR amplification, which was performed with an Eppendorf Mastercycler (Eppendorf AG, Hamburg, Germany) using primers given in Table 4 [60].

The amplification protocol was as follows: 15 min at 95 °C, followed by 35 cycles of 60 s denaturation at 95 °C, 30 s annealing at 52 °C (TVV1 and TVV3) or 54 °C (TVV2 and TVV4), and 90 s extension at 72 °C. Some 2% agarose gels stained with GelRed^®^ Nucleic Acid Gel Stain (Biotium, Inc., Hayward, CA, USA) were used to visualise the PCR products in a Gel Doc^TM^ XR+ Imager (Bio-Rad Laboratories, Inc., Hercules, CA, USA). Additionally, bands in the expected size (see Table 4) were cut out from the gel and purified with the Illustra^TM^ GFX^TM^ PCR DNA and Gel Purification Kit (GE Healthcare, Buckinghamshire, UK). Amplified products were Sanger-sequenced with Thermo Fisher Scientific SeqStudio (Thermo Fisher Scientific, Waltham, MA, USA) and compared to reference sequences in GenBank to verify the specificity of the used primers.

### 4.4. Detection of M. hominis

The QIAGEN DNA Mini Kit (QIAGEN GmbH, Hilden, Germany) was used for extraction of total DNA from the TV cultures; the DNA was stored at −20 °C for further use. Primers listed in Table 5 were used for the amplification of a *Mycoplasma* genus specific 16S rRNA gene [61]; 2% agarose gel was used to visualise the PCR products. Bands of the expected size were cut out from the gel, purified, and sequenced as described above. Reference sequences were deposited in GenBank under accession numbers OP623449–OP623451.

### 4.5. Determination of Minimal Inhibitory Concentrations (MICs)

For metronidazole susceptibility screening, 5 × 10^5^ TV cells/mL (with and without symbionts) were incubated with increasing metronidazole (Sigma-Aldrich, M3761) concentrations (0.5–512 µg/mL) in 96-well flat-bottom plates under microaerobic and anaerobic conditions in parallel at 37 °C for 48 h. To create anaerobic conditions, the microtiter plates were incubated in plastic containers with lids and equipped with one anaerobic bag (AnaeroeGen 2.5 l bag, Thermo Scientific) per 1.5 l plastic container. Viability of the trophozoites in the culture plate wells was assessed by visual examination of motile cells with an inverted microscope at 400× magnification; the lowest drug concentration where no motile TV was observed was defined as minimal inhibitory concentration (MIC). To determine the minimal lethal concentration (MLC), TV cells from the first well with no motility observed were transferred to TYM medium and cultivated for one week to confirm death of the cells.

### 4.6. LDH Assay

For the determination of cytotoxicity, the LDH assay (CyQUANT^TM^ LDH Cytotoxicity Assay Kit reaction mixture, Thermo Fisher Scientific-Life Technologies), based on the release of lactate dehydrogenase (LDH) upon cell lysis was used. Four groups of TV isolates (five isolates per group) were investigated: the first group consisted of isolates without symbionts; the second group consisted of TV with TVVs; the third group of TV with *M. hominis*; and the fourth of TV with TVVs and *M. hominis*. HeLa cells were used as target cells and cultured in IMDM Medium (Gibco, Thermo Fischer Scientific) with 10% foetal bovine serum at 5% CO_2_. The assays were performed in triplicate and repeated twice in independent set ups, and cytotoxicity was calculated according to the manufacturer’s protocol.

## Figures and Tables

**Figure 1 ijms-23-12422-f001:**
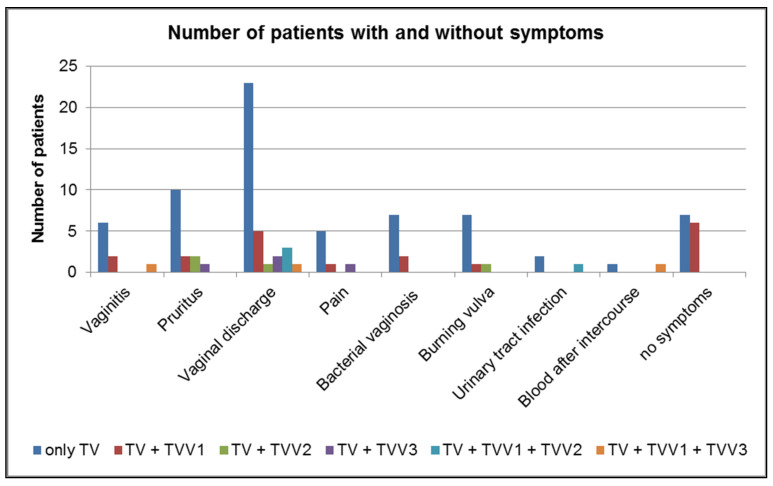
Numbers of TV infected patients without (no symptoms) or with symptoms (vaginitis, pruritus, vaginal discharge, pain, bacterial vaginosis, burning vulva, urinary tract infection, blood after intercourse) in correlation with TVV presence (TV + TVV1, TV + TVV2, TV + TVV3, TV + TVV1 + TVV2, TV + TVV1 + TVV3) or absence (only TV).

**Figure 2 ijms-23-12422-f002:**
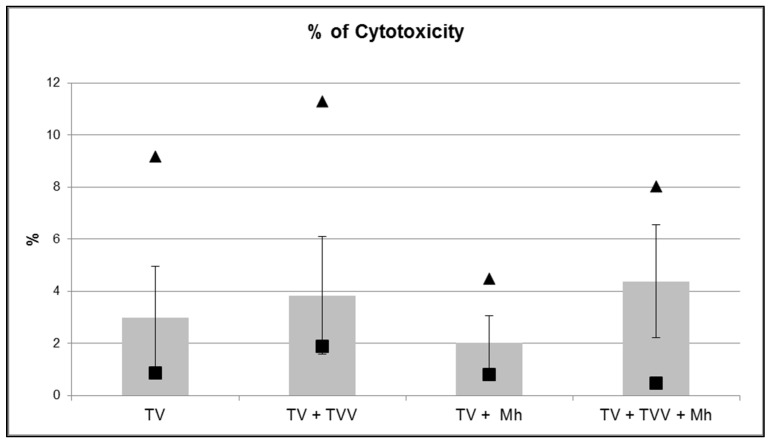
Mean percentages of *T. vaginalis* cytotoxicity of isolates without endosymbionts (TV), carrying TVVs (TV + TVV), carrying *M. hominis* (TV + Mh), and carrying TVVs and *M. hominis* (TV + TVV + Mh). Minimal (black squares) and maximal (black triangles) values of cytotoxicity are pointed.

**Table 1 ijms-23-12422-t001:** Characteristics of the study group.

Number of Patients with TV	82	
**Sex**	72 female (88%)	10 male (12%)
**Age (mean age)**	23–64 (41.2)	34–58 (46)
**No clinical symptoms reported**	7/72 (9.7%)	6/10 (60%)
**No other pathogens diagnosed**	4/72 (5.5%)	0/10 (0%)

**Table 2 ijms-23-12422-t002:** Detection of *M. hominis* in patients with TV infection and within TV isolates.

Number of Positive Patients and Cultured TV Isolates	82	
**Sex**	72 female	10 male
***M. hominis* in patients (culture)**	27/72 (36.5%)	3/10 (30%)
***M. hominis* in TV (PCR)**	20/72 (27.8%)	3/10 (30%)
***M. hominis* in patients and in TV (culture and PCR)**	10/72 (14%)	2/10 (20%)
***M. hominis* only in patients (culture), not in TV**	17/72 (24%)	1/10 (10%)
***M. hominis* only in TV (PCR), not in patients (culture)**	10/72 (14%)	1/10 (10%)

**Table 3 ijms-23-12422-t003:** Metronidazole susceptibility (mean and range) for TV harbouring different TVVs and TVV combinations, as well as TV harbouring *M. hominis* (+*M. hominis*) and TV where TVVs and *M. hominis* were present (+TVV + *M. hominis*).

TV	Metronidazole Mean Concentration (Min–Max)	
	Anaerobic Conditions	Aerobic Conditions
**no TVVs (n = 28)**	0.8 µg/mL (0.5–2 µg/mL)	3.3 µg/mL (0.5–32 µg/mL)
**+TVV1 (n = 12)**	0.65 µg/mL (0.5–1 µg/mL)	3.3 µg/mL (0.5–32 µg/mL)
**+TVV2 (n = 2)**	0.67 µg/mL (0.5–1 µg/mL)	5.5 µg/mL (0.5–8 µg/mL)
**+TVV3 (n = 5)**	0.5 µg/mL (0.5 µg/mL)	0.93 µg/mL (0.5–2 µg/mL)
**+TVV1 + TVV2 (n = 3)**	1.13 µg/mL (0.5–2 µg/mL)	4.16 µg/mL (0.5–8 µg/mL)
**+TVV1 + TVV3 (n = 2)**	1.17 µg/mL (0.5–2 µg/mL)	1.5 µg/mL (0.5–2 µg/mL)
**+*M. hominis* (n = 12)**	0.63 µg/mL (0.5–1 µg/mL)	2.58 µg/mL (0.5–32 µg/mL)
**+*M. hominis* + TVVs (n = 7)**	0.62 µg/mL (0.5–1 µg/mL)	4.65 µg/mL (0.5–32 µg/mL)

**Table 4 ijms-23-12422-t004:** Primers used for TVV determination.

*T. vaginalis* Virus	Primer Name	Primer Sequence (5′→3′)	Expected Product Size
TVV1	TVV1F2875 TVV1R3443	ATTAGCGGTGTTTGTGATGCA CTATCTTGCCATCCTGACTC	569 bp
TVV2	TVV2F2456 TVV2R3080	GCTTGAGCACTGCTCGCG TCTCTTTTGGCATCGCTT	625 bp
TVV3	TVV3F1474 TVV3R2025	TGGAGTTAAATGGTCTCGAGC GATTTCATCTTGTTCAATTGCA	552 bp
TVV4	TVV4F1338 TVV4R1834	ATGCCAGTTGCTTTCCG TTCCCCAATAGTTATCAG	514 bp

**Table 5 ijms-23-12422-t005:** Primers used for the detection of Mollicutes.

Primer Name	Target Gene	Primer Sequence (5′→3′)	Expected Product Size
GPOI MGSO	*Mycoplasma* genus specific 16S rRNA	ACTCCTACGGGAGGCAGCAGTA TGCACCATCTGTCACTCTGTTAACCTC	717 bp

## Data Availability

Not applicable.

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
