# Peer review of "Characterisation of Trichomonas vaginalis Isolates Collected from Patients in Vienna between 2019 and 2021"

_ijms, 2022, doi:10.3390/ijms232012422_

Round 1
Reviewer 1 Report
The authors describe the association between the protozoan Trichomonas vaginalis and its endosymbionts in a sample of 82 clinical isolates collected between 2019 and 2021. The work contributes to understanding the epidemiology of the association between the protozoan and its symbionts in Austria.
1. The biggest problem concerns the molecular characterization of the presence of Mycoplasma girerdii and Mycoplasma hominis, because the authors used a couple of primers that are unable to differentiate the two species. I would suggest avoiding discussion of the data for M.girerdii and only reporting the data for M.hominis. Unfortunately, M.girerdii does not grow in vitro and the results cannot be discussed, and all sentences in which M.girerdii is referred to should be deleted (e.g., lines 273-277)
2. The authors report that PCR products were cut from the gel and sequenced: however, sequence data are not reported. If these data are not discussed in the text, the sentence should be deleted
3. The authors report having studied M.hominis localized within protozoa (by detection of DNA inside T.vaginalis): in what way is the intracellular location confirmed? The use of PCR does not differentiate intracellular bacteria from membrane-associated extracellular bacteria. Were the protozoa cultured in the presence of gentamicin? Was confocal microscopy used to demonstrate intracellular localization? How can the authors rule out mixed both intracellular and extracellular localization in the same isolate?
4. A limitation of the analysis of the various isolates, both in terms of metronidazole resistance and cytopathic activity, is the important difference between different clinical isolates. The use of isogenic strains should at least be discussed as an alternative to study these aspects.
5. There are several works demonstrating the intracellularity of Mycoplasmas (e.g., M.penetrans, but also M.hominis). These papers should be cited.
6. Line 61: The intracellular location and infection capacity of M.girerdii has been previously demonstrated. The papers should be cited and the results discussed in the “Discussion” section. (see, Margarita et al. 2022 mBIO, Jun 28;13(3):e0091822. doi: 10.1128/mbio.00918-22; Aquino MF, Simoes-Barbosa A. mBIO, 2022, Aug 30;13(4):e0132322. doi: 10.1128/mbio.01323-22).
7. Results concerning the correlation between the presence of M.hominis and M.girerdii and metronidazole susceptibility reported in the paper “Margarita eV. Et al. Antibiotics. 2022 Jun 16;11(6):812. doi: 10.3390/antibiotics11060812) should be included in the text and discussed
8. The use of qPCR would allow data on multiplicity of infection (MOI). Previous data demonstrate important strain-to-strain MOI variability: these results should be discussed and reported
Author Response
A point-by-point response
Dear Editor and Reviewers!
We would like to thank you for your time and your very helpful comments and suggestions. We have carefully revised our manuscript taking into account all comments made and making appropriate changes according to the suggestions. All changes made are highlighted in the revised version of the manuscript.
Reviewer 1
The authors describe the association between the protozoan Trichomonas vaginalis and its endosymbionts in a sample of 82 clinical isolates collected between 2019 and 2021. The work contributes to understanding the epidemiology of the association between the protozoan and its symbionts in Austria.
- The biggest problem concerns the molecular characterization of the presence of Mycoplasma girerdii and Mycoplasma hominis, because the authors used a couple of primers that are unable to differentiate the two species. I would suggest avoiding discussion of the data for M.girerdii and only reporting the data for M.hominis. Unfortunately, M.girerdii does not grow in vitro and the results cannot be discussed, and all sentences in which M.girerdii is referred to should be deleted (e.g., lines 273-277)
Response: Thank you very much for your comment - following your suggestion, we have removed the data on Ca. M. girerdii and report/discuss only data concerning M. hominis. All references concerning Ca. M. girerdii (19, 20, 21, and 55) have also been deleted.
- 2. The authors report that PCR products were cut from the gel and sequenced: however, sequence data are not reported. If these data are not discussed in the text, the sentence should be deleted
Response: You are absolutely right, we did not report sequence data; however, we want to emphasize that the presence of specific bands of 569 bp, 625 bp, and 552 bp was the key confirmation for us of the presence of the TVV1, TVV2, or TVV3, respectively. For additional confirmation, purified PCR products were sent for sequencing and then compared to reference sequences in GenBank to verify the specificity of the used primers.
Primers used for the detection of M. hominis allow detection of Mycoplasma genus specific 16s rRNA gene (Ref. 61), again, in order to be sure that the organism detected was M. hominis, we sent the purified products for sequencing. Morover, as Ioannidis et al. described in 2017, also Ureaplasmas tend to be intracellular in T. vaginalis. We aimed to screen with Mycoplasma-genus specific primers in order to detect all possible intracellular/and or membrane associated Mollicutes. As shown, all sequenced PCR products have confirmed to be M. hominis.
We have now submitted three representative M. hominis sequences to GenBank. The accession number provided by GenBank is OP623449-OP623451. We added the respective sentence to “Material and Method” section.
- The authors report having studied M.hominis localized within protozoa (by detection of DNA inside T.vaginalis): in what way is the intracellular location confirmed? The use of PCR does not differentiate intracellular bacteria from membrane-associated extracellular bacteria. Were the protozoa cultured in the presence of gentamicin? Was confocal microscopy used to demonstrate intracellular localization? How can the authors rule out mixed both intracellular and extracellular localization in the same isolate?
Response: We totally agree with your comment – PCR does not differentiate intracellular bacteria from membrane-associated extracellular bacteria and we also cannot rule out a mixed localization of M. hominis within the same TV isolate. We also did not culture T. vaginalis in the presence of gentamycin. Therefore, we now explained this in the text and replaced “intracellular M. hominis” with “intracellular and/or membrane associated M. hominis”. - hominis which we detected only by PCR in TV isolates, indicated its intracellular/membrane-associated localisation, as it was not detected by culture on M. hominis special culture plates in the samples directly taken from patients.
- A limitation of the analysis of the various isolates, both in terms of metronidazole resistance and cytopathic activity, is the important difference between different clinical isolates. The use of isogenic strains should at least be discussed as an alternative to study these aspects.
Response: Thank you for pointing out the significance in the difference between the different clinical isolates. We now addressed this issue in the manuscript referring to a previous publication on this topic by our group. As described by Fürnkranz et al. (Ref. 51), differential M. hominis strains lead to different MICs in the same T. vaginalis strain. However, in this study we focused on epidemiological aspects and prevalence of endosymbionts.
- 5. There are several works demonstrating the intracellularity of Mycoplasmas (e.g., M.penetrans, but also M.hominis). These papers should be cited.
Response: Thank you for your comment, we now included more citations on this. M. hominis intracellularity was cited with reference 24, 28, 47 and 52. We mentioned also Ureaplasma intracellularity demonstrated by Ioannidis et al. in 2017 (Ref. 42).
- 6. Line 61: The intracellular location and infection capacity of M.girerdii has been previously demonstrated. The papers should be cited and the results discussed in the “Discussion” section. (see, Margarita et al. 2022 mBIO, Jun 28;13(3):e0091822. doi: 10.1128/mbio.00918-22; Aquino MF, Simoes-Barbosa A. mBIO, 2022, Aug 30;13(4):e0132322. doi: 10.1128/mbio.01323-22).
Response: As you suggested in point 1, we now deleted the section concerning Ca. M. girerdii and focus only on M. hominis in the current paper. However, we added the suggested reference (Margarita et al., 2022) (Ref. 53) in the “Discussion” section concerning M. hominis.
- Results concerning the correlation between the presence of M.hominis and M.girerdii and metronidazole susceptibility reported in the paper “Margarita eV. Et al. Antibiotics. 2022 Jun 16;11(6):812. doi: 10.3390/antibiotics11060812) should be included in the text and discussed
Response: Thank you for your suggestion, we included and discussed the suggested Reference (Ref 52) in the “Discussion” section.
The use of qPCR would allow data on multiplicity of infection (MOI). Previous data demonstrate important strain-to-strain MOI variability: these results should be discussed and reported
Response: We completely agree, qPCR would allow determining the multiplicity of infection (MOI). However, in the current study we focused on the general presence of endosymbionts and not on the MOI, which will however be taken into account in further studies. Also, currently, the determination of the MOI would only be possible for M. hominis and not for TVV, which is why we dropped this option in the current study.

Reviewer 2 Report
The manuscript written by Lesiak-Markowicz et al. is a very well written paper with many interesting aspects. The paper concerns endosymbionts in the isolates of Trichomonas vaginalis (TV) from patients with and without clinical symptoms of trichomoniasis. The authors perform many experiments including culture and PCR methods, metronidazole susceptibility and also cytotoxicity test (LDH assay). From the point of view of a parasitologist, the results are interesting and definitely worth publishing.
I have one small remark, please check the manuscript because not all the names of species are written in italics.
Author Response
A point-by-point response
Dear Editor and Reviewers!
We would like to thank you for your time and your very helpful comments and suggestions. We have carefully revised our manuscript taking into account all comments made and making appropriate changes according to the suggestions. All changes made are highlighted in the revised version of the manuscript.
Reviewer 2
The manuscript written by Lesiak-Markowicz et al. is a very well written paper with many interesting aspects. The paper concerns endosymbionts in the isolates of Trichomonas vaginalis (TV) from patients with and without clinical symptoms of trichomoniasis. The authors perform many experiments including culture and PCR methods, metronidazole susceptibility and also cytotoxicity test (LDH assay). From the point of view of a parasitologist, the results are interesting and definitely worth publishing.
I have one small remark; please check the manuscript because not all the names of species are written in italics.
Response: Response: Thank you very much for your kind comments and for pointing out the inconsistencies in the use of italics. We are very sorry for this and now carefully checked the entire manuscript including all figures and tables, all species names are now in italics. All changes are highlighted in yellow.

Round 2
Reviewer 1 Report
The manuscript can be published in the present firm